# The electric field cavity array effect of 2D nano-sieves

Fan Xu [1,4], Yuke Li [2,4], Qing Zou[1], Yu Shuang He[1], Zijia Shen[1], Chen Li[1], Huijuan Zhang[3], Feipeng Wang[1], Jian Li[1] & Yu Wang [1,3] ✉

For the upsurge of high breakdown strength ($E_b$), efficiency ($\eta$), and discharge energy density ($U_e$) of next-generation dielectrics, nanocomposites are the most promising candidates. However, the skillful regulation and application of nano-dielectrics have not been realized so far, because the mechanism of enhanced properties is still not explicitly apprehended. Here, we show that the electric field cavity array in the outer interface of nanosieve-substrate could modulate the potential distribution array and promote the flow of free charges to the hole, which works together with the intrinsic defect traps of active $Co_3O_4$ surface to trap and absorb high-energy carriers. The electric field and potential array could be regulated by the size and distribution of mesoporous in 2-dimensional nano-sieves. The poly(vinylidene fluoride-co-hexa-fluoropropylene)-based nanocomposites film exhibits an $E_b$ of 803 MV m$^{-1}$ with up to 80% enhancement, accompanied by high $U_e$ = 41.6 J cm$^{-3}$ and $\eta \approx$ 90%, outperforming the state-of-art nano-dielectrics. These findings enable deeper construction of nano-dielectrics and provide a different way to illustrate the intricate modification mechanism from macro to micro.

With the development of power system and energy system dominated by renewable energy, new requirements for advanced electrical and energy materials are continuously being put forward[1–3]. Exploiting high electric field breakdown performance ($E_b$) engineering insulating materials is of great significance for the design and manufacture of power equipment insulation system in the third-generation power grid;[2] developing high-energy-density ($U_e$) and high-energy-efficiency ($\eta$) electrostatic capacitor dielectric is prominence for reducing the volume and weight of energy storage devices[4,5]. Since the concept of nano-dielectrics is first proposed by T. J. Lewis in 1994, numerous efforts show that the incorporation of nanoparticles (NPs) could significantly improve the electrical, thermal, and mechanical properties of dielectric materials due to their nano "interface" effect is more vital than itself structure[4–17]. Therefore, nanocomposites dielectric is one of the most promising candidates for advanced functional insulation and electrostatic energy storage material.

Polymer is both an integral part of many insulating devices, such as cable, and a critical component of electrostatic energy storage devices, such as metallized film capacitor, which is widely applied in electrical and energy territory[18–20]. Poly(vinylidene fluoride-co-hexa-fluoropropylene) (P(VDF-HFP)) has the advantages of high dielectric constant (K), stable structure, corrosion resistance, and heat resistance, etc., and is a promising competitor as insulating energy storage material[21,22]. So, it is adopted as the substrate in this experiment.

Polymer-based nano-dielectric is a research hotspot in the film capacitor field in the past decade and quite a few efforts report excellent modified performance[13,18–25]. However, the modification mechanism is still not been understandably clarified, resulting in us not being able to regulate it maturely and realize rational construction of

[1]State Key Laboratory of Power Transmission Equipment & System Security and New Technology, and the School of Electrical Engineering, Chongqing University, 174 Shazheng Street, Shapingba District, 400044 Chongqing City, P. R. China. [2]Department of Chemistry and Centre for Scientific Modeling and Computation, Chinese University of Hong Kong, Shatin, Hong Kong, P. R. China. [3]The School of Chemistry and Chemical Engineering, Chongqing University, 174 Shazheng Street, Shapingba District, 400044 Chongqing City, P. R. China. [4]These authors contributed equally: Fan Xu, Yuke Li. ✉e-mail: wangy@cqu.edu.cn

the nanocomposites. Trap theory is well recognized as a way to depict the essential features of charge carrier transport, with shallow traps assisting charge transport and deep traps holding charge for long-lasting time[10,26]. The depth of the trap is determined by the location of the trap energy level between the conduction and valence band, depending on the natural defect of the material itself. It is a micro-level theoretical explanation[27]. We call these kinds of traps as "intrinsic charge traps" (eV) in this paper. But, the interconnection from microscopic charge trapping and de-trapping to macroscopic free charge carriers' transport and dielectric local discharge is not well investigated since the failure process is complicated. The application of density functional theory (DFT)[28] and finite element simulation (FES)[29] benefited from advanced computer technology provides the possibility to build a "bridge" between them for the illustration of the mechanism, while little attention has been paid to this aspect in the past. Meanwhile, most of the papers on these two technologies are sometimes too coarse or simplified to embody the experimental results precisely. For example, the high external electric field ($E_{ex}$) application context of the insulating materials is rarely considered in the DFT, and the dealing of interface layer parameters is unrefined in FES. Experimentally, most of the nanocomposites are carried out on the basis of NPs, and some works on 1D and 2D nanomaterials have emerged in recent years[13,30,31]. Most nanofillers are coarse or purchased from the market. After simple treatment, such as surface decoration, etc., they are incorporated into the insulating substrate. 2D nanomaterial is a rising star due to its huge specific surface, and unique electrical, thermal, optical, and electromagnetic properties[13,32]. But the lack of nanomaterials' synthesis and regulation techniques leads to the scarcity of systematic structural research, resulting in a disconnected understanding of the modification mechanism.

In this work, we synthesize homogeneous nano-sieves of different pore sizes and incorporate them uniformly into P(VDF-HFP)-based films with low doping. It is found that the DC breakdown field improvement has never been so high, and by some distance. Such a large breakdown electric field is achieved. Theoretical calculations reveal that the electric field cavity array and sub-macro potential trap array in the outer interface layer allure the free-charged ions or electrons toward the holes of nano-sieves, making them effectively captured by the deep intrinsic defect traps on the active $Co_3O_4$ surface. Meanwhile, the dielectric loss is suppressed, even comparable to that of linear dielectrics. The close-knit connection ($F-Co^{3+}$) between nano-sieves (large specific surface area) and polymers significantly affects the polarization of P(VDF-HFP).

## Results

### Characterization of nanofillers and nanocomposite films

$Co_3O_4$ is a p-type antiferroelectric semiconductor with multiple metal valence states[33], and is adopted as the additive. Uniformly belt-like $Co_3O_4$ nano-sieves (NSIs) with four different pore sizes (5, 15, 25, 50 nm) are synthesized by regulating hydrothermal reaction and calcination conditions (Fig. 1a, Supplementary Note 1, and Methods section). It can be seen from the scanning electron microscope (SEM) images (Fig. 1a, Supplementary Fig. 1 and 2) that the NSIs have a quasi-one-dimensional (1D) nanobelt morphology, with a width of 200–400 nm and a length of 2–3 um. High-resolution transmission electron microscope (HRTEM) image (Fig. 1b) and fast Fourier transform pattern (inset) show the preferential exposure of (110) facet of spinel $Co_3O_4$. In addition, NPs (16 nm) are obtained from our previous work for controlled experiments (Supplementary Fig. 3). At the same test conditions, the XRD peak of $Co_3O_4$ NPs is higher and narrower than that of NSIs, showing that NPs have better crystallinity while NSIs have a higher specific surface area and more defects[34]. In order to quantify the specific surface area, nitrogen adsorption and desorption isotherms based on Brunauer−Emmett−Teller (BET) theory are tested. As shown in Supplementary Fig. 5, the specific surface area of NSIs is

generally ~385.5 m2/g, which is 5.7 times larger than that of NPs (67.6 $m^2$/g). These results show that we have successfully synthesized NSIs with large specific surface area and distinct pore sizes.

P(VDF-HFP) (10 mol% HFP) with higher K of 9−10 at 1 kHz is selected as the matrix to investigate the nano-modification mechanism and performance. Using the serial dilution method to obtain low doping concentration (0.0008, 0.008, 0.08 wt%) nano-suspensions and the classic casting method to fabricate thin films (Fig. 1c, Supplementary Note 2). The fabricated films are highly flexible and translucent. In the case of 0.0008 and 0.008 wt% doping, equal to "impurity" level content, the naked eye cannot distinguish any difference from the pure film; the film turns black gradually as the further increase of nanofillers (Fig. 1c, and Supplementary Fig. 6). Each nanocomposites film is uniquely labeled as X@NSIs-Y wt% (X denoted as pore size, and Y denoted as doping amount). The surface SEM images of films intuitively reflect their crystalline state (Supplementary Fig. 7). Pure P(VDF-HFP) film has a spherulite structure (Supplementary Fig. 7a). After the addition of NSIs, the surface crystalline state of the film presents fibrous and smooth, which indicates NSIs might affect the nucleation of growth process and lead to the increment of β-phase content (Supplementary Fig. 7b–d)[35]. But when NSIs are excessive, it would hinder the growth of crystal domains (smaller grains), resulting in the increase of α-phase content (Supplementary Fig. 7e, f). Front high-angle annular dark-field scanning TEM (HAADF-STEM) and cross-sectional SEM images of 15@NSIs-3wt% film display that NSIs are randomly and uniformly distributed in the ~18 um thick film, and no agglomeration occurred (Fig. 1d and Supplementary Fig. 8). It can be seen from HRTEM images of 15@NSIs-3wt% (Fig. 1e) and pure film (Supplementary Fig. 9) that P(VDF-HFP)-based films contain poly-crystalline and amorphous regions, and NSIs are in direct contact with the amorphism region. Fourier-transform infrared (FTIR) and XRD techniques are employed to characterize the α, β, and γ phases of the films (Fig. 1f, and Supplementary Figs. 11−15). The results before and after heat treatment, the pure film's β content decreased by 16% (42.3 → 35.65%), while that of 15@NSIs-0.008 wt% only reduced by 8% (47.3 → 43.5%) (Supplementary Fig. 11), illustrates that NSIs facilitate the formation of β-phase during polymer crystal growth, and inhibit the conversion of β to α during heat treatment[36]. With the increase of the doping mass fraction, the β-phase content increases first and then decreases (Supplementary Table 1), which is consistent with the previous film's surface SEM images (Supplementary Fig. 7). In a typical 15@NSIs-0.008 wt% film, the relative content of α, β, and γ phases is 20.1%, 43.5%, and 36.4%, respectively (Supplementary Fig. 14). The nanocomposites films with different pore sizes have nearly identical XRD and FTIR results at the same doping mass, which shows that the pore size has little effect on the phases component of P(VDF-HFP)-based film (Supplementary Fig. 15, and Supplementary Table 1).

### Electrical properties

The electrical properties of P(VDF-HFP) and its nanocomposites with $Co_3O_4$ nanomaterials are estimated at room temperature (Fig. 2, Supplementary Note 3). The dielectric spectrums as a function of frequency are measured (Fig. 2a, Supplementary Figs. 16−17). In the case of "impurity-level" doping (0.0008 and 0.008 wt%), the introduction of $Co_3O_4$ NSIs with higher K ($\varepsilon_r$, ~20)[37] actually leads to a decreased K of the nanocomposites film compared to pure film, which is inconsistent with conventional cognition. When the filler fraction is further increased, the K value is mounting and exceeding that of pure film (Fig. 2a). The spectrum of loss factor and conductivity has similar variation pattern, which first decreases and then increases with the rising of doping concentration (Supplementary Fig. 16). The nanocomposites with different pore sizes NSIs have almost the same dielectric spectrum at 0.008 wt% (Supplementary Fig. 17). Combined with aforementioned XRD and FTIR results (Supplementary Fig. 15), it can be deduced that pore size has little influence on nanocomposites'

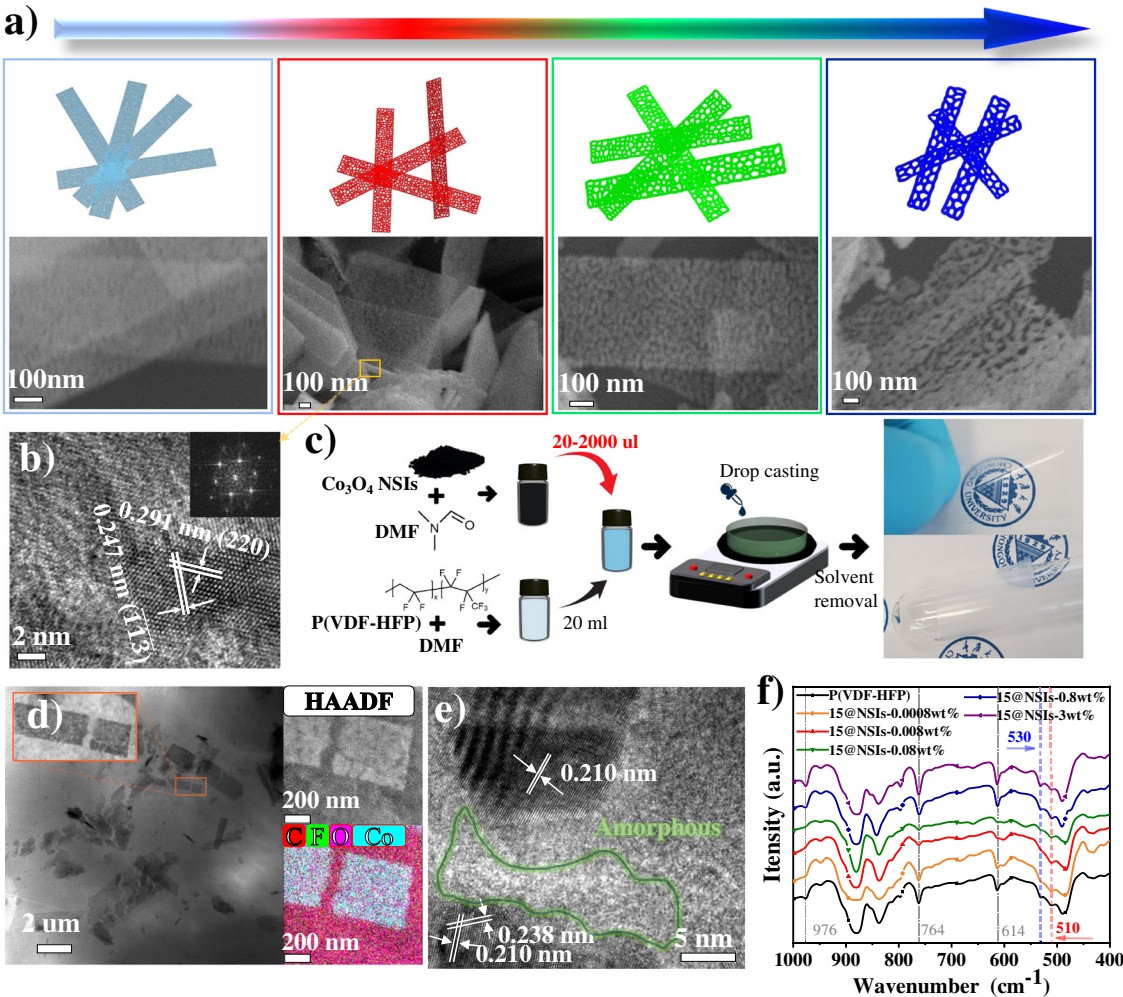

**Fig. 1 | Material preparation and characterization. a** Diagram (top) and SEM (bottom) images of the $Co_3O_4$ nano-sieves (NSIs) with 5, 15, 25, and 50 nm pore diameter (left to right). **b** HRTEM image of NSIs with 15 nm pore size. The lattice fringe spacings of 0.291 and 0.247 nm correspond to the (220) and ($\overline{1}\overline{1}3$) crystal planes, respectively, indicating that the ($1\overline{1}0$) and (110) zone axes have normal direction perpendicular to the crystal planes. So, the preferentially exposed crystal face is (110) in accordance with the FFT pattern (inset). **c** Schematic and photographs of the preparation of P(VDF-HFP)-based nanocomposites films (~20 μm-thick) doped with 0.008 wt% $Co_3O_4$ NSIs (15 nm pore size) (denoted as 15@NSIs-0.008wt%). N,N-Dimethylformamide (DMF). **d** Front HAADF-STEM image of 15@NSIs-3wt%. Ar ion beam is used to thin the film before tests. **e** HRTEM of 15@NSIs-3wt%. **f** FTIR spectra of pure P(VDF-HFP) film, 15@NSIs-0.0008 wt%, 15@NSIs-0.008 wt%, 15@NSIs-0.08 wt%, 15@NSIs-0.8 wt%, and 15@NSIs-3wt%.

properties films at low voltage. The two-parameters Weibull plots are employed for statistical analysis of the breakdown field strength ($E_b$) data (Fig. 2b, Supplementary Figs. 18–20). Impressively, the $E_b$ of 15@NSIs-0.008 wt% is 803.3 MV m$^{-1}$, which is 40% higher than for the NPs/P(VDF-HFP)-0.008 wt% (575 MV m$^{-1}$), >80% higher than for pure film (~445 MV m$^{-1}$). The elevation magnitude is superior to state-of-the-art reported works (Fig. 2b,c, Supplementary Table 2). The $Co_3O_4$ NSIs have various modification states to $E_b$ when the filler fraction is less than 3 wt%, which also contributes to high-energy storage performance. Here, the pore size of NSIs exhibits obviously diversity in $E_b$ (Supplementary Figs. 19, and 20). It can be clearly seen that 15@NSIs-0.008 wt% has higher $E_b$ than other pore sizes-based films. The pore size has significant domination on breakdown strength which is a sharp contrast to previously obtained conclusion at low voltage, indicating that there are other key factors are crucial to the breakdown process in high electric field.

We next evaluate the energy storage properties of the films from their unipolar electric displacement-electric field (D-E) loops under room temperature (Fig. 2d–f, Supplementary Note 4). The discharge energy density ($U_e$) reaches a maximum of 41.6 J cm$^{-3}$ for 15@NSIs-0.008 wt% at 800 MV m$^{-1}$ because of the excellent dielectric strength

(Fig. 2d). The value is 3.4 times as large as the 12.2 J cm$^{-3}$ for pure P(VDF-HFP) film. An overall upgraded energy efficiency (η, ~90%) is discovered, which is superior to the great majority of electrostatic energy storage capacitors (Fig. 2e)[9,11,15]. As shown in Fig. 2d, a set of concentration control experiments, the D-E loops of 15@NSIs-0.0008 wt% and 15@NSIs-0.008 wt% are located below that of pure film, corroborated with aforementioned reduced K, suggesting the interaction between the NSIs and macromolecule inhibits partial polymer polarization. While the NSIs are overdosed, the loop shifts to the upper left with increasing loss, which is probably due to the polarization action of $Co_3O_4$ NSIs itself structure far exceeding the inhibition effect of the interface[38]. The polarization of 15@NSs-3wt% continues to increase after the maximum electric field is reached, creating a bloated shape of the D-E loop. This demonstrates that significant leakage currents are present in 15@NSs-3wt%[39]. All of those merits demonstrate that the P(VDF-HFP)-based film modified by the nano-sieves with suitable pore size at low doping is a promising candidate for electrostatic storage application because of combined great $U_e$ and high η, which surpasses the bulk P(VDF-HFP)-based film and nano-modified dielectrics (Fig. 2f, Supplementary Table 2)[4,40–48].

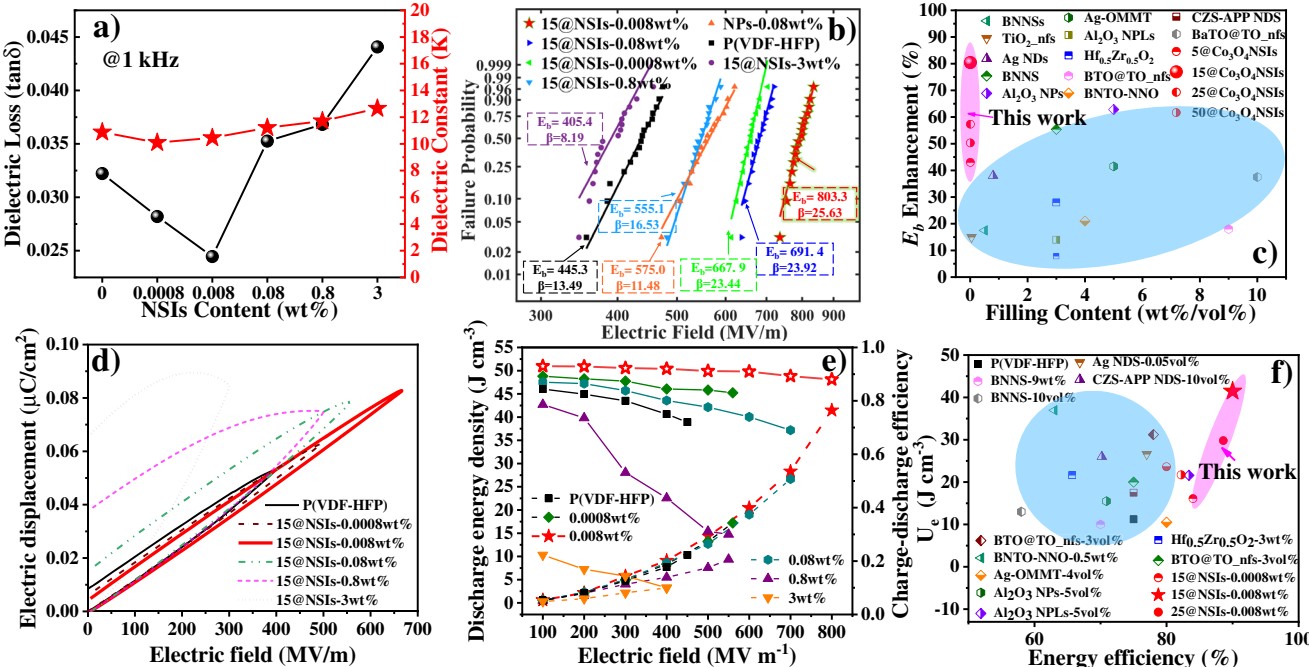

**Fig. 2 | Dielectric properties and energy storage capability. a** Relative permittivity and dispassion factor of P(VDF-HFP) and Co₃O₄ NSIs-based films as a function of NSIs content. **b** Two-parameter Weibull distribution of the breakdown strength of the P(VDF-HFP) and 15@NSIs/P(VDF-HFP) with different filler contents; the characteristic breakdown strength $E_b$ and shape parameter $\beta$ are derived from each composition. **c** Comparison of breakdown strength enhancement ratio as a function of filler content between this work and other literatures. **d** Unipolar D-E loops of P(VDF-HFP) and 15@NSIs/P(VDF-HFP) with different filler mass fraction at 50 Hz. **e** Energy density and efficiency values of the P(VDF-HFP) and 15@NSIs-0.008 wt% with respect to applied electric fields up to their breakdown fields. **f** Comparison of the discharge energy density between this work and other state-of-the-art reported dielectric film and nanocomposites.

## Density functional theory

To figure out why is that the NSIs modified film has lower polarization loss and higher $E_b$, the interfacial molecular model of Co₃O₄ and P(VDF-HFP) is established for first-principles calculation (Fig. 3a–h). Nowadays, most of the studies on DFT so far are carried out under simplified vacuum background without considering actual operational environment for insulation dielectrics materials that are long-term exposed to high $E_{ex}$, which might lead to inaccurate results[4]. So, DFT is used to study the polymer adsorption on the Co₃O₄(110) surface with the electric field effects. In the Fig. 3a, Co²⁺ and Co³⁺ are in the tetrahedral and octahedral oxygen surrounding, respectively. Co²⁺ and Co³⁺ are in the top layer of the stable Co₃O₄(110) in the Fig. 3c and d. To simplify the polymer and Co₃O₄(110) surface interaction, only one unit molecule as shown in Fig. 3b is used for the adsorption study. The F atom is the electron donor comparing with H and C atom. We first find the binding site at 0.00 eV/Å. The molecule binds with Co²⁺ with F-Co bond distance 2.18 Å in Fig. 3e, f. For the Co³⁺ binding, the F atom of the molecule is shared by two Co³⁺ with F-Co distance 2.602 and 2.885 Å (Fig. 3g, h). The adsorption energy ($E_{ads}$) are −0.79 and −1.08 eV with F atom binding with Co²⁺ and Co³⁺, respectively. Generally, the shorter binding distance, the stronger binding. In our case, the molecule prefers to bind with Co³⁺. We speculate there are two reasons, one is the positive charge on metal atom, the other is unsaturated coordination. The positive charge on the Co³⁺ is of course bigger than the Co²⁺. The coordination number of Co²⁺ and Co³⁺ are 4 and 6 in the bulk, respectively. Once, in the top layer of surface, the coordination number of Co²⁺ and Co³⁺ change to 3 and 4. The internal electric field is formed with a large electrostatic potential difference ΔE 12.33 eV in Fig. 3i, signifying the active charge transfer[4]. The disordered macromolecular chain in the amorphous phase of the P(VDF-HFP) film is more susceptible to steering polarization, which is the main part of polarization loss[49]. It is obtained that nanofillers are in direct contact with the amorphous polymer molecular chains from the previous characterization. The formation of F-Co³⁺ bond results in enhanced binding between nanofiller and macromolecule (» the van der Waals force). This allows the orientation polarization loss to be well suppressed, which well accounted for the introduction of high K inorganic nanofillers at low doping content leading to the reduction of polarization losses. Supplementary Table 4 demonstrates that there are electrons transfer (7.3 μC/m²) from Co₃O₄(110) surface to the adjoining polymer molecular layer (denoted as inner interface layer) when the $E_{ex}$ = 0 MV/m. An increase in the charges on the polymer molecules results in enhanced molecular polarization. However, there are differences in charge transfer between them under different intensities and orientations of $E_{ex}$, which leads to various magnitudes of molecular polarization in the inner interface layer (Fig. 3j). Based on the DFT calculation results, we revise the interface model in the next FES work. Furthermore, there are abundant deficiencies such as unsaturated/broken bonds on the surface of the Co₃O₄ NSIs. If the new energy levels introduced by these defects exist in the energy gap between valence band and conduction band, it will trap electrons or holes. Shallow traps are formed near the bottom of the conduction band or the top of the valence band[27]. As illustrated in Supplementary Fig. 23, the surface of Co₃O₄ nanomaterials contains both deep and shallow traps.

## Finite element simulation

The above analysis provides a well interpretation for the suppressed polarization loss of the nanocomposites, but it is still ambiguous why the 15@NSIs-0.008 wt% film has the highest $E_b$. It is generally accepted that the charge carrier transport and E distortion are essential factors in the performance of insulating materials[50]. So, we simulate the steady-state internal E and potential (V) distribution of the interfacial corrected dielectric model by using finite element computations (Fig. 4, and Supplementary Note 6). The potential contours and E distribution in the selected vertical cross-section surface at the NSIs centrum (Supplementary Fig. 26) are plotted in P(VDF-HFP) substrate

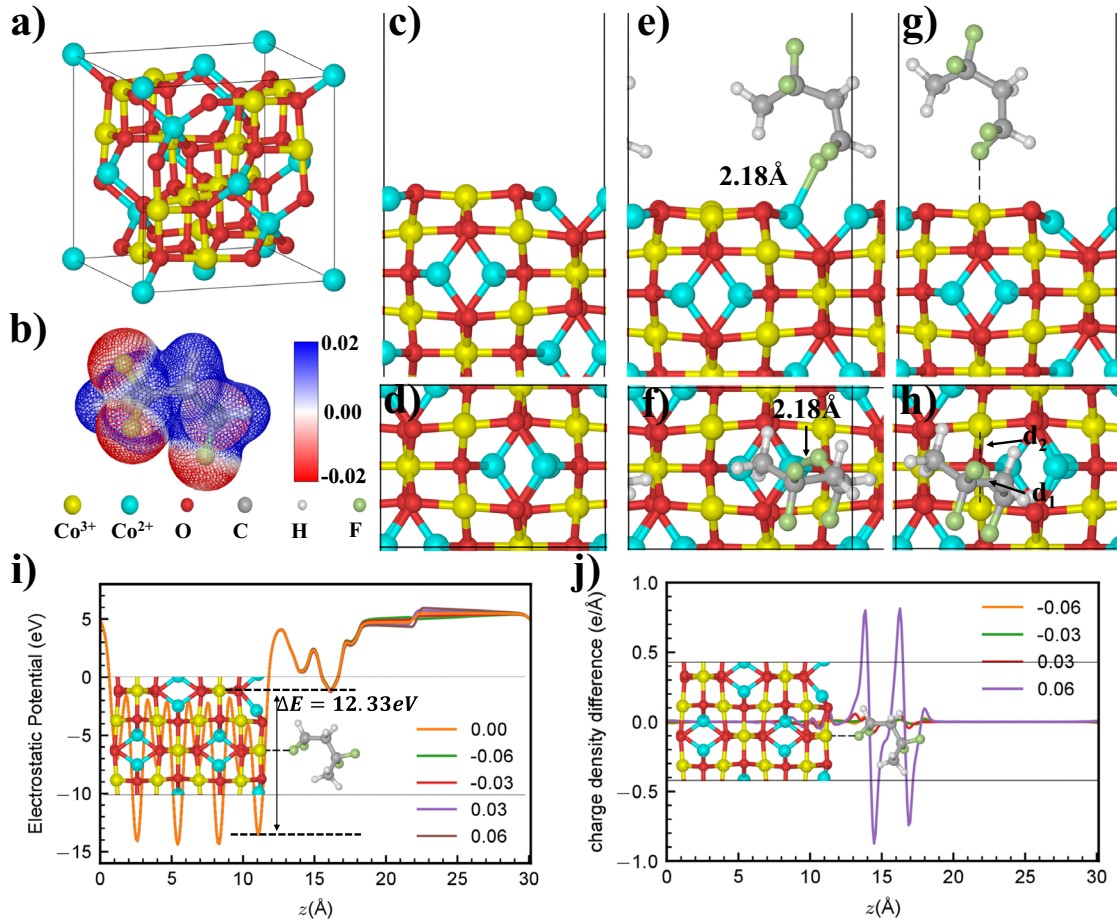

**Fig. 3 | Density functional theory (DFT) study on the molecule adsorption on Co3O4(110) surface taking the electric filed into consideration. a** The bulk structure of Co₃O₄. It contains Co²⁺ and Co³⁺. **b** The electrostatic potential surface (ESP) of the molecule. **c,d** side and top view of the Co₃O₄ (110) surface with the Co³⁺, Co²⁺ and O atoms exposure. **e, f** Side and top view of the molecule adsorption on Co²⁺ site. **g, h** Side and top view of the molecule adsorption on the Co³⁺ site. **i** The electrostatic potential along $z$-axis at electric field 0.00 eV/Å with value ΔE 12.33 eV.

The reason for the uneven electrostatic potential curve in the 20−30 Å vacuum area is that Co₃O₄ surface is adsorbed, resulting in a dipole in the whole slab. **j** The charge density difference along $z$-axis of molecule adsorption on Co³⁺ with −0.06, −0.03, 0.03 and 0.06 eV/Å (the positive direction is left → right). The charge density difference is the charge density in a $E_{ex}$ minus the charge density in 0.00 eV/Å. The molecule polarization of polymer is enhanced when the $E_{ex}$ is positive, and vice versa.

excited by a $E_{ex}$ = 200 MV m⁻¹. Compared to the nanosheet and nano-particle, it can be seen that the equipotential lines close to nano-sieve boundary are meandering due to the existence of mesoporous, which indicates the V and E distributions are intricate in this area (Fig. 4a, b, Supplementary Figs. 25b and 27). The non-uniform E and V distribution could produce crucial electric stress on charges and change their transport paths. In order to visually understand the potential trap and E distribution near NSIs surface, the normE and V data of local hor-izontal plane (0.5 nm from the inner interface layer, Supplementary Fig. 28) are extracted and three-dimensional (3D) plotted. As shown in Fig. 4d, the E in the region matched up with the sieve pores is wea-kened, while the E in the surface region corresponding to the Co₃O₄ skeleton is enhanced, resulting in a 3D tubular-like cavity array of electric field. The E cavity distribution makes the potential in the outer interface layer present a sub-macroscopic potential distribution array (Fig. 4e). The E cavity effect is effective in the range of 0–10 nm around the interface (Supplementary Figs. 30, and 31). When a positive ion transport from positive to negative electrode, the ion would move toward the pore of nano-sieves, acting like a trap. The same is true for negative charges. To distinguish it from the intrinsic defect trap, the potential distribution caused by uneven E are called "sub-macro potential trap array" (V). A series of comparative 3D graphs of normE and V of NPs, nanosheets, and nano-sieves with different pore sizes are shown in Supplementary Fig. 25. The15@NSIs in polymer substrate

excited by external E generates the largest and most sophisticated interface area with alternating configuration of suitable deep and shallow potential traps array, which might be the critical element why the 15@NSIs-0.008 wt% has the highest $E_b$ performance. Relative potential difference ($V_r$) and electric field distortion rate (ξ) are employed to provide a quantitative description of potential traps depth and E deformity, respectively. The $V_r$ and ξ generated by the nanoparticle are 0.31 V and 40%. The $V_r$ generated by the 2D nano-materials is reduced to ~0.21 V, but the ξ value is decreased by half. $V_r$ and ξ are negatively correlated. What we pursue is more potential traps and lower ξ. At this point, the emergence of 2D porous sieve nano-materials could satisfy these two demands. This could be the critical factor for the world-leading insulation performance modification of 15@NSIs-0.008 wt% in this experiment.

## Discussion

In this paper, homogeneous Co₃O₄ nano-sieves (NSIs) with different pore sizes and NPs are synthesized by hydrothermal method, and well-dispersed P(VDF-HFP)-based nanocomposites films with trace doping concentration are fabricated by serial dilution and casting methods. The most impressive electrical performance ($E_b$ = 803.3 MV m⁻¹, η = 90%, and $U_e$ = 41.6 J cm⁻³) is achieved by the nanocompo-sites film with 15 nm pore size nano-sieves at 0.008 wt% doping amount (denoted as 15@NSIs-0.008 wt%). The 80% dielectric strength

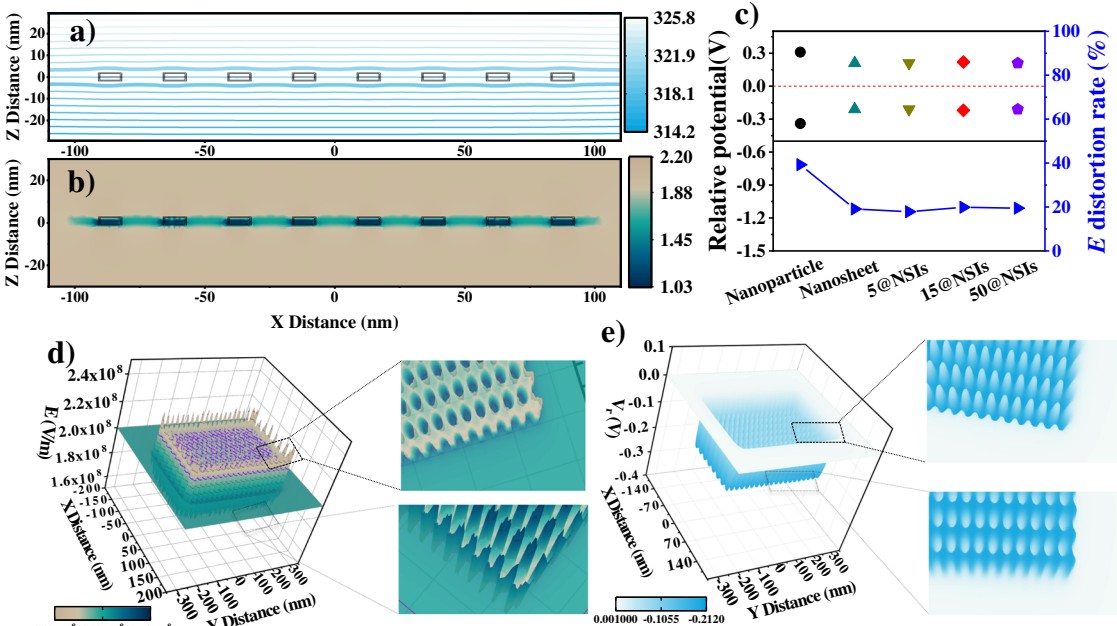

**Fig. 4 | Steady-state E and V distribution. a** Equipotential lines around the nano-sieve in P(VDF-HFP) dielectric. **b** Local electric field distortion (vertical direction) near a nano-sieve/polymer matrix interface under an external E = 200 MV/m. **c** Relative potential difference ($V_r$) and electric field distortion rate ($\xi$) induced by, nanosheet, 5@NSIs, 15@NSIs, and 50@NSIs. ($V_r = V_{min} - V_{theory}$, $\xi = ((E_{max} - E_{ex})/E_{ex}) \times 100\%$; where $V_{min}$ is minimum of V in the selected area, $V_{theory}$ is the theoretical potential without nanomaterials at the site, $E_{max}$ is maximum of E in the selected area, $E_{ex}$ is

externally applied electric field. **d** 3D graphs of $V_r$ of 15@NSIs-0.008 wt% in the selected surface (the image in the upper right is a visual enlargement from the top right to the bottom, the image in the lower right is a visual enlargement from the bottom right to the top). **e** 3D graphs of normE of 15@NSIs−0.008 wt% in the selected surface (the image in the upper right is a visual enlargement from the top to the bottom, the image in the lower right is a visual enlargement from the bottom to the top).

improvement is 2.7 times that of NPs-0.008 wt% film, which surpasses almost state of art reported nano-modified dielectrics. But what is more noticeable is that its η is close to 90% at any electric field, comparable with linear dielectric, which is superior to pure film (75%) and most other products in its class. First-principles calculation suggests the existence of charge transfer between F and $Co^{3+}$, restraining the orientation polarization of linked macromolecules, which could be well accounted for the decreased polarization losses and K of the composites film after introducing high K nanofillers. The $E_{ex}$ context is considered into insulating dielectric DFT for the first time, and it is found that the polymer chain attached to the $Co_3O_4$ (denoted as inner interface layer) receives electrons from the surface of nanomaterial to enhance the molecular polarization. While the charge transfer between them is influenced by $E_{ex}$ resulting in divergence polarization at different interface sites. Based on this, we divide the inner interface layer in FES model into upper and lower layers and revise their parameters. The simulation demonstrates that the electric field (E) cavity array caused by the 2D nano-sieve structure and material difference produces a micron-scale non-uniform potential distribution array in the outer interface layer. This potential array plays a trap-like role in acceleration or buffering free charge carriers, causing the carriers to move toward the sieve holes, named as "sub-macro potential trap array". Impressively, the sub-macro potential trap array could be tuned by the size and distribution of mesoporous in 2D nanomaterials. This might be the rational interpretation for the nano-sieves doped film achieving such a remarkable modification effect.

The E cavity array and sub-macro potential trap array is induced by nano-sieves in the outer interface layer under high $E_{ex}$ modulating the free carriers' transport. The combination of sub-macro potential trap array with intrinsic defect traps on $Co_3O_4$ active surface diminishes high-speed moving ions or electrons in insulating dielectric, hinders the discharge channels' formation, hugely improves the breakdown resistance of the nanocomposite films. Meanwhile, the

polarization losses are suppressed since the depressed macromolecule orientation polarization in the inner interface layer and the large specific surface area of nano-sieves. The 2D nanomaterial structure design that we developed to improve overall dielectric energy storage performance should be applicable to other material systems and nanocomposites. It may also be helpful for other attempts to elucidate the modification mechanism.

## Methods

### Reagents
Ethylene glycol (EG) (99%, Adamas,), aqueous ammonia ($NH_3 \cdot H_2O$,) (29%, Adamas), cobalt nitrate hexahydrate ($Co(NO_3)_2 \cdot 6H_2O$) (99.99%, Sigma-Aldrich), sodium carbonate ($Na_2CO_3$) (99.8%, Fisher), N,N-Dimethylformamide (DMF) (99.8%, J.T.Baker), Poly(vinylidene fluoride-co-hexafluoropropylene) (P(VDF-HFP)) (Kynar Flex 2801, 10%). All the reagents are used as received. Deionized water (-18.2 MΩ cm⁻¹) is employed throughout the study.

### Synthesis of 2D Co₃O₄ nano-sieves with different pore sizes
In a typical preparation of $Co_3O_4$ nano-sieves with a mean pore size of 15 nm, denote as 15@$Co_3O_4$ NSIs, 12.5 ml $NH_3 \cdot H_2O$ and 15 ml EG are evenly mixed in a 50 ml beaker, then 3 ml 1 M $Na_2CO_3$ solution is added and form homogeneous solution after 2 min of stirring. Afterwards, 5 ml 1 M aqueous cobalt nitride solution is added to the above mixed solution stirring for 20 min. The resulting homogeneous purple blends are transferred into a 50 ml Teflon-lined stainless steel autoclave for sealing and fixation, which is then heated in a temperature-preset oven at 170 °C for 1020 min. After the autoclave naturally cools down to room temperature, the precipitation is centrifuged and rinsed at least 4 times by deionized water and then once with ethanol. Finally, the products are dried in a 60 °C of vacuum oven overnight and calcined in a program-controlled muffle furnace at 450 °C with a ramp rate of 1 °C/min.

The amounts of precursors and calcination temperature required for the synthesis of $Co_3O_4$ nano-sieves with other pore sizes and morphology are as follows.

5@$Co_3O_4$ NSIs: 12.5 ml $NH_3 \cdot H_2O$, 16 ml EG, 2 ml 1 M $Na_2CO_3$ and 5 ml 1 M $Co(NO_3)_2$, 250 °C.

25@$Co_3O_4$ NSIs: 12.5 ml $NH_3 \cdot H_2O$, 12 ml EG, 5 ml 1 M $Na_2CO_3$ and 5 ml 1 M $Co(NO_3)_2$, 350 °C.

50@$Co_3O_4$ NSIs: 12.5 ml $NH_3 \cdot H_2O$, 11 ml EG, 7 ml 1 M $Na_2CO_3$ and 5 ml 1 M $Co(NO_3)_2$, 550 °C.

$Co_3O_4$ nanoparticles: 12.5 ml $NH_3 \cdot H_2O$, 12.5 ml deionized water, and 5 ml 1 M $Co(NO_3)_2$, 250 °C.

## Film sample fabrication

The classic solution-casting method is employed in this study to prepare dielectric film. To make low-concentration nanocomposites film, 1 g of P(VDF-HFP) powers are firstly dissolved into 20 ml of DMF solvent with vigorous stirring for 2 h to obtain a homogeneous transparent solution A. 8 mg of $Co_3O_4$ nanomaterial is dispersed in 20 ml of DMF: acetone = 1:1 solvent in a 20 ml glass bottle, under sonicated for 2 h and stirred 0.5 h to form a uniform suspension B. Following that, a certain amount of X (X = 20, 200, and 2000 ul) of B is added into solution A to obtain a mixed solution with different concentrations of $Co_3O_4$ nanomaterial, corresponding to 0.0008 wt %, 0.008 wt%, and 0.08 wt% respectively. After sonication for 2 h and stirring for 12 h, the mixture is cast onto a clean and flat quartz glass plate (cleaned with isopropanol and ethanol (1:1)) and dried at 40 °C thermostat for 3 h. Afterwards, the glass plate is transferred to a 120 °C vacuum oven for 8 h to allow the solvent to evaporate completely. Finally, the flexible film is peeled off after soaking in ice water for 3 min. Film thickness is controlled by concentration of P(VDF-HFP). The thickness of all the films used for electrical tests is 15–20 um.

Pure P(VDF-HFP) and high-concentration nanocomposites films are fabricated with identical processes, except that the method of obtaining the mixed solution before casting is different. For 0.8 wt% and 3 wt% doping concentration, 8 mg and 30 mg of the $Co_3O_4$ nanomaterial are weighed and added directly to the A solution. For pure film, solution A could be used directly.

## Characterization

The morphology and structure of nano-fillers and films are determined by field emission scanning electron microscopy (FSEM, JEOL JSM-7800F, 10 kV), X-ray diffraction (XRD, PANalytical X'Pert Powder with Cu Kα radiation), transmission electron microscopy (TEM, FEI, Talos F200S, 200 kV). X-ray photoelectron spectroscopy (XPS, ESCA-LAB250) and BET surface area measurement (BET, Quantachrome Autosorb-2MP) are adopted to analyze the surface element valence and surface area and pore size of $Co_3O_4$ nano-sieves. Fourier-transform infrared spectra (FTIR, Nicolet iS50) is performed from 400 to 4000 $cm^{-1}$. Differential scanning calorimetry (DSC, Netzsch. Ltd DSC 404 F3) is conducted at a heating/cooling rate of 10 °C $min^{-1}$. Thermogravimetric analysis (TGA, MERRLER TOLEDO TGA2) test is carried out with a heating rate 10 °C $min^{-1}$ from ~30 °C to 700 °C under nitrogen atmosphere.

## Electrical measurement

Gold electrodes of diameter 2 mm and thickness of 50 nm are sputtered on both sides of the polymer films by using a high vacuum coater (Leika EM ACE 600). Frequency-dependent dielectric spectras over the frequency range between 0.1 Hz and 1 MHz at room temperature are collected with a Concept 80 broadband dielectric spectrometer (Novocontrol Technologies GmbH & Co. KG, Hundsangen, Germany). High-field electric displacement-electric field (D-E) loops are measured using a ferroelectric test system (aixACCT-TF Analyzer 2000) with a triangular unipolar wave under 50 Hz at room temperature. Dielectric breakdown strength tests are carried out with DC voltage at a ramping rate of 200 V $s^{-1}$ and a limiting current of 5 mA using a KEYSIGHT 33500B as the signal generator and a TREK PO621P as voltage amplifier.

## Data availability

The data that support the findings of this study are available from the corresponding author upon request. The data and code are available on request or specifying the conditions of access. Source data are provided as a Source data file. Source data are provided with this paper.

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

## Acknowledgements

This work was financially supported by the National Natural Science Foundation of China (NSFC, Grant No. U19A20100, 21971027), Thousand Young Talents Program of the Chinese Central Government (Grant No. 0220002102003) and Project 111 (BP0820005).

## Author contributions

Y.W. and F.X. conceived the concept and devised the experimental program; F.X., Y. L., and Y. H. performed research; F.X. and C. L. analyzed data; all co-authors commented on the paper, F.X. and Y. W. wrote the paper.

## Competing interests

The authors declare no competing interests.
