## [Peer Review File · Nature Communications]

The electric field cavity array effect of 2D nano-sievesREVIEWER COMMENTS

Reviewer #1 (Remarks to the Author):

The authors reported homogeneous nano-sieves of different pore sizes and incorporate them uniformly into P(VDF-HFP)-based films with extremely low doping. This paper could be accepted for publication after minor corrections;

- 1- The novelty of the present work should be discussed in more details.
- 2- The authors should provide mapping of the elements with the SEM images.
- 3- The interaction of Co₃O₄ with polymer matrix should be confirmed from XRD and FTIR data.

Reviewer #2 (Remarks to the Author):

The manuscript reports the interesting results that Co₃O₄ nano-sieves achieve a substantial amelioration in the dielectric strength and energy storage properties of P(VDF-HFP)-based films through an extremely low doping ratio. The nano-sieves' peculiar structure is considered to induce the electric field cavity array effect under the action of external voltage, which causes an array of alternating deep and shallow potential traps. The high energy free charge carriers are induced and trapped by the array and intrinsic deep traps on the surface of Co₃O₄ nano-sieves, thereby realizing a significant improvement in breakdown voltage. I'd like to suggest publication in NC after addressing the following items.

1. Cobalt is harmful. Why choose Co₃O₄ nano-sieves as the additive? How about other candidates?
2. The nano-sieves generate an electric field cavity effect under the action of external voltage, which regulates the local electric field distribution and guides the transport and capture of charge carriers. How the range for the cavity effect around the nano-sieves?
3. Provide the temperature for the D-E tests. How is the possible nonlinear characteristic of the sample films, e.g. higher temperatures? Explain why the maximum polarization of sample 15@NSs-3wt% does not correlate to the highest electric field in Figure 2d?
4. The DFT is used to correct the interface parameters in FES with the dielectric constant variation of the upper and lower inner interface layers. Supply the theory basis for correcting the dielectric constant. Define it is a qualitative or quantitative correction route.
5. In Figure 3i, why the electrostatic potential curve is not flat when there is no external electric field (E_{ex}) in 20~30Å vacuum area? The intensity of the unevenness increases with the positive E_{ex} applied, while it seems to decrease with the negative E_{ex} applied. Need to supply a reasonable explanation.

Response Letter to Reviewers' Comments

November 21, 2022

Dear Reviewers,

We sincerely appreciate the valuable time the reviewers have spent reviewing our manuscript and providing insightful comments and suggestions to help further improve the quality of our work. Considering the reviewers' evaluations, we have made a point-by-point response to the reviewers' comments and revised our manuscript to improve the clarity of our work. We believe we have addressed all of the reviewers' comments and now the paper is more rigorous in content and clearer in presentation. Our point-by-point responses to the reviewers' comments are as follows.

Looking forward to hearing from you,

Sincerely,

Yu Wang

COMMENTS TO AUTHOR:

Reviewer #1:

The authors reported homogeneous nano-sieves of different pore sizes and incorporate them uniformly into P(VDF-HFP)-based films with extremely low doping. This paper could be accepted for publication after minor corrections.

1- The novelty of the present work should be discussed in more details.

[Answers]

We are grateful for the reviewer's comments. Nanocomposite dielectric is a research hotspot in the high-voltage insulation field in the past decade and quite a few efforts report excellent modified performance. Most of the works are carried out based on nanoparticles, and some works on 1D and 2D nanomaterials have emerged in recent years. But most nanofillers are coarse or purchased from the market. After simple treatment, such as surface decoration, etc., they are incorporated into the insulating substrate. The lack of nanomaterials synthesis and regulation techniques leads to the scarcity of systematic structural research, resulting in a disconnected understanding of the modification mechanism. Microscopically, intrinsic defect trap theory is well recognized as a way to depict the essential features of charge carrier transport. However, the interconnection from the microscopic charge trapping to macroscopic free charge carriers' transport and dielectric local discharge is not well investigated. The skillful regulation and application of nano-dielectrics are greatly hindered.

The work in this paper is at the forefront of interdisciplinary. Utilizing the mature Co_3O_4 fabrication techniques, the nano-sieves structure is elaborately constructed and applied to the P(VDF-HFP) matrix through steady optimization. Such a large breakdown electric field is achieved with extremely low doping. The DC breakdown field improvement has never been so high, and by some distance. Combined with DFT and FES calculations, the electric field cavity array effect induced by the 2D nano-sieve on the macro scale is revealed, and an alternately distributed potential array is formed. This macro potential array plays a trap-like role in accelerating or buffering free charge carriers, causing the carriers to move toward the sieve holes, named as "sub-macro potential trap array". The combination of sub-macro potential trap array with intrinsic defect traps on the Co_3O_4 active surface diminishes high-speed moving ions or electrons in the dielectric. Here, we have revised the corresponding discussions and explanations in the text.

Page 1, line 28: "These findings enable deeper construction of nano-dielectrics and provide a novel way to illustrate the intricate modification mechanism from macro to micro."

Page 2, line 26: "Most nanofillers are coarse or purchased from the market. After simple treatment, such as surface decoration, etc., they are incorporated into the insulating substrate. 2D nanomaterial is a rising star due to its huge specific surface, and unique electrical, thermal, optical, and electromagnetic properties.^{13,32} But the lack of nanomaterials' synthesis and regulation techniques leads to the scarcity of systematic structural research, resulting in a disconnected understanding of the modification mechanism."

In this work, we synthesize homogeneous nano-sieves of different pore sizes and incorporate them uniformly into P(VDF-HFP)-based films with extremely low doping. It is found that the DC breakdown field improvement has never been so high, and by some distance. Such a large breakdown electric field is achieved with extremely low doping. Theoretical calculations reveal that the electric field cavity array and sub-macro potential trap array in the outer interface layer allure the free-charged ions or electrons toward the holes of nano-sieves, making them effectively captured by the deep intrinsic defect traps on the active Co_3O_4 surface. Meanwhile, the dielectric loss is amazingly suppressed, even comparable to that of linear dielectrics. The close-knit connection (F-Co^{3+}) between nano-sieves (large specific surface area) and polymers significantly affects the polarization of P(VDF-HFP).”

2- The authors should provide mapping of the elements with the SEM images.

[Answers]

Thanks for the questions raised by the reviewer, this is very helpful for the integrity of the paper. We are sorry for ignoring to present the detailed mapping of the elements in the previous submission. There is only a small mixed elements mapping in Figure 1d (bottom right-hand corner), and the icon is too small to present clearly. Here, Figure 1d is modified and the mappings of elements (C, F, O, and Co) are complemented in supporting information (Figure S10).

In Figure S10, the C and F elements are evenly distributed throughout the field of the window. The O is also widely distributed in the substrate, which is paradoxical since the P(VDF-HFP) does not contain oxygen element. This is because the operation of Ar ion beam milling to the film before TEM, while the high surface temperature makes the oxidation of C. But the Co element is only distributed on the skeleton surface of the nano-sieve. These demonstrate that nano-sieves are randomly and uniformly distributed in the P(VDF-HFP) substrate, and no agglomeration occurred.

Figure 1d. Front HAADF-STEM image of 15@NSIs-3wt%. Ar ion beam is used to thin the film before tests. e, HRTEM of 15@NSIs-3wt%.

Figure S10. (a) HAADF-STEM image of 15@NSIs-3wt%. HAADF-STEM-EDX elemental mapping images showing the element distributions of b) C, c) F, d) O and e) Co. The carbon is oxidized during the operation of Ar ion beam milling.

3- The interaction of Co_3O_4 with polymer matrix should be confirmed from XRD and FTIR data.

[Answers]

We thank the reviewer for the constructive comments on our work and also for pointing out the important insight to emphasize the results. From the HRTEM of 15@NSIs-3wt% (Figure 1e), it can be obtained that the nano-sieve is in direct contact with the P(VDF-HFP)'s amorphous. XRD is sensitive to crystalline information and retarded to the non-crystal structure. So, the conventional XRD test analysis we adopt can only reflect the crystalline phases of the P(VDF-HFP)-based film macroscopically. FTIR spectroscopy reflects the type of functional groups contained in a substance and the chemical environment in which it is located, and the phase composition of a polymer film could be determined by analysis of the characteristic spectrum and fingerprint region. However, to obtain the atomic-level interactions between Co_3O_4 and polymer molecules for our low-doped nanocomposite films, at least the current state-of-the-art high-energy in-situ characterizations are required to make it possible. This kind of method is still in the exploration stage. Therefore, in this paper, conventional XRD and FTIR are employed to characterize the nanocomposite films.

As seen in Figure 1f and Figure S12-14, the α -phase content of the P(VDF-HFP)-based samples first decreases and then increases with increasing doping amount. At low concentration doping ($< 0.008\text{wt}\%$), Co_3O_4 nano-sieves are beneficial to the nucleation of macromolecules, leading to the increase of β -phase. But when Co_3O_4 overdoses, it would hinder the growth of crystal domains

(smaller grains), resulting in the increase of α -phase (Table S1). The XRD patterns show a 36.8° characteristic peak of Co_3O_4 in Figure S12 (15@NSIs-0.8wt% and 15@NSIs-3wt%). Here, we have revised corresponding discussions and explanations in the text (Page 3, line 35) and Supporting Information Section 2.

Page 3, line 35: “Fourier-transform infrared (FTIR) and XRD techniques are employed to characterize the α , β , and γ phases of the films (Fig. 1f, and Fig. S11-15). The results before and after heat treatment, the pure film’s β content decreased by 16% (42.3→35.65 %), while that of 15@NSIs-0.008wt% only reduced by 8% (47.3→43.5 %) (Fig. S11), illustrates that NSIs facilitate the formation of β -phase during polymer crystal growth, and inhibit the conversion of β to α during heat treatment.³⁶ With the increase of the doping mass fraction, the β -phase content increases first and then decreases (Table S1), which is consistent with the previous film’s surface SEM images (Fig. S7). In a typical 15@NSIs-0.008wt% film, the relative content of α , β , and γ phases is 20.1%, 43.5%, and 36.4%, respectively (Fig. S14). The nanocomposites films with different pore sizes have nearly identical XRD and FTIR results at the same doping mass, which shows that the pore size has little effect on the phases component of P(VDF-HFP)-based film (Fig. S15, and Table S1)”

Figure 1 | Material preparation and characterization. a, Diagram (top) and SEM (bottom) images of the Co_3O_4 nano-sieves (NSIs) with 5, 15, 25, and 50 nm pore diameters (left to right). b, HRTEM image of NSIs with 15 nm pore size. The lattice fringe spacings of 0.291 and 0.247 nm correspond to the (220) and $(\bar{1}\bar{1}\bar{3})$ crystal planes,

respectively, indicating that the ($\bar{1}10$) and (110) zone axes have normal direction perpendicular to the crystal planes. So, the preferentially exposed crystal face is (110) in accordance with the FFT pattern (inset). **c**, Schematic and photographs of the preparation of P(VDF-HFP)-based nanocomposites films (~20 μm -thick) doped with 0.008wt% Co_3O_4 NSIs (15 nm pore size) (denoted as 15@NSIs-0.008wt%). **d**, Front HAADF-STEM image of 15@NSIs-3wt%. Ar ion beam is used to thin the film before tests. **e**, HRTEM of 15@NSIs-3wt%. **f**, FTIR spectra of pure P(VDF-HFP) film, 15@NSIs-0.0008wt%, 15@NSIs-0.008wt%, 15@NSIs-0.08wt%, 15@NSIs-0.8wt%, and 15@NSIs-3wt%.

Supporting Information Section 2:

Figure S12. XRD patterns of pure P(VDF-HFP) film, 15@NSIs-0.0008wt%, 15@NSIs-0.008wt%, 15@NSIs-0.08wt%, 15@NSIs-0.8wt%, and 15@NSIs-3wt%. $2\theta=36.8^\circ$ is Co_3O_4 characteristic peak.

Figure S13. Histogram of β content-doping fraction.

Figure S14. The relative content and scheme illustration of α -, β -, and γ -phase in 15@NSIs-0.008wt%. the sphere colored by black, red, and light blue represent carbon, fluorine and hydrogen atoms, respectively.

Reviewer #2:

The manuscript reports the interesting results that Co_3O_4 nano-sieves achieve a substantial amelioration in the dielectric strength and energy storage properties of P(VDF-HFP)-based films through an extremely low doping ratio. The nano-sieves' peculiar structure is considered to induce the dielectric filed cavity array effect under the action of external voltage, which causes an array of alternating deep and shallow potential traps. The high energy free charge carriers are induced and trapped by the array and intrinsic deep traps on the surface of Co_3O_4 nano-sieves, thereby realizing a significant improvement in breakdown voltage. I'd like to suggest publication in NC after addressing the following items.

- 1. Cobalt is harmful. Why choose Co_3O_4 nano-sieves as the additive? How about other candidates?**

[Answers]

Thank you very much for this insightful comment. Indeed, cobalt is a heavy metal, and excessive intake can induce various toxic effects on the body. Moreover, it will also generate great harm to the natural environment, such as water pollution. But cobalt oxide is a multifunctional, antiferromagnetic p-type semiconductor (with a direct optical bandgap of 1.48 and 2.19 eV) that has been used in electrochromic sensors, energy storage, heterogeneous catalysis, pigments, dyes, and in lithium-ion rechargeable batteries as an anode material.¹ Because of their interesting physical properties, cobalt oxide also has spintronic applications. The pollution could be inhibited if we pay attention to leakage and recovery in industrial and engineering applications.

- (1) Why choose Co_3O_4 nano-sieves as the additive?

Thank you for pointing this out. Both Co and Fe are transition metals of Group VIII, and their oxides have similar physical properties in many respects. Fe_3O_4 is proven to be an effective dielectric additive. Co_3O_4 is a p-type antiferroelectric semiconductor with multiple metal valence states. It has higher activity than Fe_3O_4 in the field of electrochemistry and catalysis, which might be because the Co_3O_4 active surface has more defects such as unsaturated coordination bonds.² These defects can act as deep intrinsic trap sources in the context of the high electric field in dielectric insulating materials, playing a key role in the modulation of high-energy free carriers. So, we believe that Co_3O_4 nanomaterials may produce superior modification effects. In our previous work, we have demonstrated that Co_3O_4 nanomaterials as an additive have a good modification effect in transformer oil.³ In order to systematically study the role of nano-additives' structure in dielectrics, Co_3O_4 is selected as the additive in the paper.

Page 2, line 44: " **Co_3O_4 is a p-type antiferroelectric semiconductor with multiple metal valence states³³, and is adopted as the additive.**"

- (2) How about other candidates?

Thank you for the recommendation. In this paper, Co_3O_4 is adopted as a tool to explore the structure effect of nanomaterials in nanocomposites dielectric. It is obtained that the electric field cavity array effect of 2D nano-sieves is well beneficial to irregulate the transport and capture of charges. The

types of nano-additives still need systematic exploration and optimization. Co_3O_4 could be replaced by other semiconductor metal oxides, such as Fe_3O_4 (environmentally friendly) or Mn_3O_4 (more metal mid-price states). The filler could also be substituted with conductive or insulating nanomaterials. The difference in dielectric constant between the substrate and the filler is a double-edged sword that requires exhaustive consideration. The larger the difference between them, the greater the induced local electric field distortion; if the difference is too small, the electric field cavity array may not play a role.

- [1] Raman, V. et al. Synthesis of Co_3O_4 nanoparticles with block and sphere morphology, and investigation into the influence of morphology on biological toxicity. *Experimental and Therapeutic Medicine* **11**, 553-560 (2016).
- [2] Saputra, E. et al. A comparative study of spinel structured Mn_3O_4 , Co_3O_4 and Fe_3O_4 nanoparticles in catalytic oxidation of phenolic contaminants in aqueous solutions. *Journal of colloid and interface science* **407**, 467-473 (2013).
- [3] Xu, F. et al. Seeking optimized transformer oil-based nanofluids by investigation of the modification mechanism of nano-dielectrics. *Journal of Materials Chemistry C* **8**(22), 7336-7343 (2020).

2. The nano-sieves generate an electric field cavity effect under the action of external voltage, which regulates the local electric field distribution and guides the transport and capture of charge carriers. How the range for the cavity effect around the nano-sieves?

[Answers]

Thanks for the questions raised by the reviewer, this is very helpful for the rigor and integrity of the paper. We further conducted finite element simulation to figure out the potential and electric field distribution around the interface. We have revised the corresponding discussions and explanations in the text (Page 6, line 10). The simulation results are presented in Figure S30 and Figure S31.

In the region far from the interface (>20 nm), the planar potential distribution exhibits a relatively weaker and wider pit trap (Figure S30a-b). Its effect is similar to that of nanosheets. However, in the region of 0~10 nm, the planar potential shows an intricate 3D trap array distribution, reflecting the electric field cavity effect caused by the porous structure of the nano-sieve (Figure S30c-e). As the distance increases, the depth of the potential trap decreases, showing an exponential distribution (Figure S31). Therefore, the electric field cavity effect is effective in the range of 0~10 nm around the interface.

Page 6, line 10: “**The E cavity effect is effective in the range of 0~10 nm around the interface (Figure S30, and 31).**”

100 nm

50 nm

5 nm

1 nm

0.5 nm

Distance from interface

Figure S30. The planar potential distribution 3D graphs of V_r of 15@NSIs near the interface (a) 100nm, (b) 50 nm, (c) 5nm, (d) 1 nm, and (e) 0.5 nm. In the region far from the interface (>20 nm), the planar potential distribution exhibits a relatively weaker and wider pit trap. Its effect is similar to that of nanosheets. However, in the region of 0~10 nm, the planar potential shows a intricate 3D trap array distribution, reflecting the electric field cavity effect caused by the porous structure of the nano-sieve.

Figure S31. The depth of potential trap (V_r) - the distance from the interface curve.

- 3. Provide the temperature for the D-E tests. How is the possible nonlinear characteristic of the sample films, e.g. higher temperatures? Explain why the maximum polarization of sample 15@NSs-3wt% does not correlate to the highest electric field in Figure 2d?**

[Answers]

Thank you for the valuable professional advice given by the reviewers. We apologize for not indicating the temperature for the D-E test in the text. During ferroelectric testing, all operations are performed at room temperature. Here, we have revised and underlined the test temperature in the text (Page 4, line 24).

Page 4, line 24: “We next evaluate the energy storage properties of the films from their unipolar electric displacement-electric field (D-E) loops under room temperature (Fig. 2d-f, SI Section 4).”

- (1) How is the possible nonlinear characteristic of the sample films, e.g. higher temperatures?

Thanks for the professional comments made by the referee, which have greatly inspired us. At room temperature, it is found that the interaction between Co_3O_4 nano-sieves and P(VDF-HFP) inhibits partial polymer polarization. The unipolar electric displacement-electric field (D-E) loop of 15@NSIs-0.008wt% film exhibits smaller losses and higher efficiency, even comparable to the linear dielectric. However, for P(VDF-HFP), there is a significant difference in the unipolar D-E loop at various temperatures. The higher the temperature, the higher the energy and activity of the polymer molecule, the higher the loss, and the more pronounced the nonlinear characteristics.¹ As a result, at higher environment or application temperatures, it could be inferred that the introduction

of an appropriate amount of Co_3O_4 nano-sieves would also suppress the polarization losses of the P(VDF-HFP)-based film to some extent, but the degree of the nanocomposites' nonlinearity would be strengthened.

- (2) Explain why the maximum polarization of sample 15@NSs-3wt% does not correlate to the highest electric field in Figure 2d?

We are grateful for the reviewer's comments. Indeed, the polarization of the unipolar loop of 15@NSs-3wt% continues to increase after the maximum electric field is reached, creating a bloated shape of the D-E loop. This phenomenon is not only seen in this paper, but also occurs in some literatures¹⁻³. This demonstrated that significant leakage currents are present in 15@NSs-3wt%. Here, we have revised the corresponding discussions and explanations in the text (Page 4, line 35).

Page 4, line 35: "The polarization of 15@NSs-3wt% continues to increase after the maximum electric field is reached, creating a bloated shape of the D-E loop. This demonstrates that significant leakage currents are present in 15@NSs-3wt%."³⁹

- [1] Zhou, L. et al. Enhancing thermal stability of P (VDF-HFP) based nanocomposites with core-shell fillers for energy storage applications. *Composites Science and Technology* **186**, 107934 (2020).
- [2] Walker, J. et al. Electric field dependent polarization switching of tetramethylammonium bromotrichloroferrate (III) ferroelectric plastic crystals. *Applied Physics Letters* **116**(24), 242902 (2020).
- [3] Yan, H. et al. The contribution of electrical conductivity, dielectric permittivity and domain switching in ferroelectric hysteresis loops. *Journal of Advanced Dielectrics* **1**(01), 107-118 (2011).

- 4. The DFT is used to correct the interface parameters in FES with the dielectric constant variation of the upper and lower inner interface layers. Supply the theory basis for correcting the dielectric constant. Define it is a qualitative or quantitative correction route.**

[Answers]

We deeply appreciate the time and effort you have spent in carefully reviewing our manuscript. It is a qualitative correction route. Bader charge analysis is employed to analyze the charge transfer of the system before and after adsorption. There is a charge transfer (F-Co^{3+}) between the polymer and C_3O_4 molecule, and its degree is influenced by the external electric field (E_{ex}) (Figure S22, and Table S4). The positive E_{ex} is defined to be vertically downward. A positive E_{ex} deepens the charge transfer, and vice versa weakens it. For polymer, molecular polarization is enhanced when more electrons are transferred from Co_3O_4 to the macromolecular. So, we believe that the dielectric constant of polymer molecules in the inner interface layer is increased when a positive E_{ex} is applied. For the sieve-polymer interface, there are always two sides, one in the same direction as E_{ex} and the other in the opposite direction. Therefore, we divide the inner interface layer into upper and lower

interfaces in the FES. The dielectric constants of these two interfaces are set to be different. This is a rough qualitative correction. It is found that this interface difference has little effect on the conclusions of this paper. So, there is no further exploration in this respect. But it provides a simple idea for combining DFT and FES. In future work, it could be quantitatively analyzed through rigorous mathematical derivation and comprehensive consideration. Here, we have revised corresponding discussions and explanations in Supporting Information Section 5 and 6.

Supporting Information Section 5:

Table S4. Interface charge transfer under different E_{ex} . ($\text{Co}_3\text{O}_4 \rightarrow \text{P}(\text{VDF-HFP})$ molecular is positive direction)

Electric field (eV/ Å)	Charge transfer ($\mu\text{C}/\text{m}^2$)	Charge transfer ratio (%)
0.00	0.0073	0
-0.06	0.0066	-9.59
-0.03	0.0070	-4.11
0.03	0.0077	5.48
0.06	0.0083	13.70

Figure S22. The calculated charge density difference between polymer and Co_3O_4 .

Supporting Information Section 6: “In contrast to the routine COMSOL electric field simulation of nanoparticles-modified dielectrics that only consider two parts, fillers and substrates, does not in-depth consider about the interface region, which is generally recognized to be the part that plays a key role in enhancing the electrical performance of the dielectrics. Thus, a shell layer 0.5 nm thick is coated on the surface of the nanofillers to represent the interfacial region where the polarization of the polymer molecule is affected according to the results of first-principles calculations in the section 5. The K of upper interface layer (UIL) is set to 14, the K of lower interface layer (LIL) is set to 10. This is a rough qualitative correction. As shown in Fig. S25a and b, it is well-observed that the electric potential lines are more intensive on the upper and lower sides of nanoparticles because of the mis-match in K values of the fillers and polymer matrix, which results in the concentrated electric field on the filler-polymer interfaces. In the simulation result (Fig. S25c)

without interfacial layer, the E distortion on the upper and lower surfaces caused by nanoparticles is symmetric. But, in the simulation plot (Fig. S25d) with interfacial layer correction, the E distortion on the upper and lower surfaces caused by nanoparticles is asymmetric. The enhanced polymer molecular polarization of UIL achieves to a better buffer matching in filler-polymer, which results in a reduced degree of E distortion. This might be essential to achieve increased breakdown strength of dielectric nanocomposites. Therefore, the breakdown mechanism of nano-modified dielectrics could be explicated more reasonably by using interfacial layer correction method.”

- 5. In figure 3i, why the electrostatic potential curve is not flat when there is no external electric field (E_{ex}) in 20~30Å vacuum area? The intensity of the unevenness increases with the positive E_{ex} applied, while it seems to decrease with the negative E_{ex} applied. Need to supply a reasonable explanation.**

[Answers]

We thank the reviewer for the constructive comments on our work and also for pointing out the important insight to emphasize the results. Dipole correction is considered in the DFT calculation. In VASP software, LDIPOL=. TRUE. Parameter must be added to consider the electric field. The thickness of the vacuum layer is 12Å from the position of 20Å to 32Å, which is the range of the vacuum layer usually considered in the DFT calculation. The reason for the uneven electrostatic potential curve is that the Co_3O_4 surface is adsorbed, resulting in a dipole in the whole slab. Relevant posts include an introduction and discussion.^{1,2} The external electric field affects this dipole polarization, so it leads to a different flatness of the electrostatic potential curves. Here, we have revised the corresponding discussions and explanations in the caption of Figure 3i.

Figure 3 | Density functional theory (DFT) study on the molecule adsorption on $\text{Co}_3\text{O}_4(110)$ surface taking the electric field into consideration. **a**, The bulk structure of Co_3O_4 . It contains Co^{2+} and Co^{3+} . **b**, The electrostatic potential surface (ESP) of the molecule. **c-d**, side and top view of the $\text{Co}_3\text{O}_4(110)$ surface with the Co^{3+} , Co^{2+} and O atoms exposure. **e-f**, Side and top view of the molecule adsorption on Co^{2+} site. **g-h**, Side and top view of the molecule adsorption on the Co^{3+} site. **i**, The electrostatic potential along z axis at electric field $0.00\text{eV}/\text{\AA}$ with value $\Delta E = 12.33\text{eV}$. The reason for the uneven electrostatic potential curve in $20\sim 30\text{\AA}$ vacuum area is that Co_3O_4 surface is adsorbed, resulting in a dipole in the whole slab. **j**, The charge density difference along z axis of molecule adsorption on Co^{3+} with $-0.06, -0.03, 0.03$ and $0.06\text{ eV}/\text{\AA}$ (the positive direction is left \rightarrow right). The charge density difference is the charge density in a E_{ex} minus the charge density in $0.00\text{ eV}/\text{\AA}$. The molecule polarization of polymer is enhanced when the E_{ex} is positive, and vice versa.

[1]<https://ionizing.page/post/vasp-dipol-correction-work-function/>

[2]https://www.researchgate.net/post/Why_VASP_gives_no_straight_vacuum_line_for_the_surface_lo cal_potential

REVIEWERS' COMMENTS

Reviewer #1 (Remarks to the Author):

Accepted for publication in the present form

Reviewer #2 (Remarks to the Author):

All questions have been addressed well by the authors, and in this stage it could be considered for acceptance.